# *In vitro* activity of MRS-2541, a novel MetRS inhibitor, against a selection of resistant gram-positive organisms associated with serious hospital infections

John Domagala,[1] Seyedhameneh Jahanbakhsh,[2] Angela K. Mendez,[2] Chris M. Pillar,[2] Andrea Marra,[2] David A. Hufnagel,[2] Nora M. R. Molasky,[3] Zhongsheng Zhang,[4] Erkang Fan,[4] Dawn Reyna,[1] Elke Lipka,[1] Frederick S. Buckner[3]

**ABSTRACT** New antibacterials are still needed to treat serious gram-positive hospital infections, including acute bacterial skin and skin structure infections and bloodstream infections. MRS-2541 is a novel type 1 methionyl-tRNA synthetase inhibitor highly selective for gram-positive bacteria. MRS-2541 was evaluated with multiple comparator antibiotics against a large panel of gram-positive organisms enriched with antibacterial-resistant phenotypes. MRS-2541 was the most potent agent against 45 *Staphylococcus aureus* (SA) (6 SAs, 27 methicillin-resistant *Staphylococcus aureus*, and 12 vancomycin-intermediate *S. aureus*) and 35 coagulase-negative staphylococci with $MIC_{90}$ values of 0.06 and 0.12 µg/mL, respectively, which was superior to all comparators, including vancomycin, linezolid, and daptomycin. MRS-2541 was very potent against 15 *Enterococcus faecalis* (4-VRE) and 13/15 *Enterococcus faecium* at ≤0.016 µg/mL, with two insensitive *faecium* strains (>16 µg/mL). MRS-2541 was less potent against *Streptococcus* spp., with $MIC_{90}$ values vs *S. pyogenes*, *S. pneumoniae*, and *S. sanguinis* of 0.5 µg/mL. *S. agalactiae* and the viridans group streptococci were much less sensitive, with $MIC_{90}$ values of 8 to >16 µg/mL. These data support MRS-2541 as a new potential option for serious gram-positive hospital infections.

**IMPORTANCE** Infections caused by antibiotic-resistant bacteria are inherently challenging to treat. Poor outcomes, including higher death rates, are associated with infections caused by antibiotic-resistant organisms. As a result, there is an urgent need for new antibiotics, particularly ones that act via novel mechanisms that evade established resistance mechanisms. In this paper, we provide new data about MRS-2541, a preclinical antibiotic that acts by inhibiting the bacterial methionyl-tRNA synthetase. We show that MRS-2541 has potent in vitro activity against diverse strains of *Staphylococcus*, *Streptococcus*, and *Enterococcus*. For example, all 115 strains of *Staphylococcus* spp. (including 27 methicillin-resistant *Staphylococcus aureus* strains and 12 vancomycin-intermediate *S. aureus* strains) were fully susceptible to MRS-2541 with MICs of <0.25 µg/mL. The specific activity against gram-positive bacteria and sparing of gram-negative flora indicates that MRS-2541 may be less disruptive of intestinal microbiota than broad-spectrum antibiotics. MRS-2541 is a promising preclinical candidate for infections due to resistant gram-positive organisms.

**KEYWORDS** antibiotics, gram-positive, *Staphylococcus*, *Streptococcus*, *Enterococcus*, tRNA synthetase, MetRS inhibitors

Globally, there are over 55 million acute bacterial skin and skin structure infections (ABSSSIs) per year (1). In the United States, ABSSSIs account for over 14 million healthcare visits per year, with 4 million emergency department visits and over 900,000

Address correspondence to Frederick S. Buckner, fbuckner@uw.edu.

The authors declare no conflict of interest.

See the funding table on p. 9.

yearly hospital admissions (1, 2). The incidence continues to rise with the increasing aged population due to multiple risk factors such as diabetes, COPD, and prior exposure to healthcare facilities (2–4). Moreover, the incidence of recurrent infection is 26%, and the all-cause mortality rate in 2019 was 3.6% (5), which places an enormous burden on the healthcare system of over $14 billion (2). ABSSSIs are caused almost exclusively by staphylococci and streptococci, with *Staphylococcus aureus* (SA) and *Streptococcus pyogenes* being the most common pathogens (2, 6). Methicillin-resistant *Staphylococcus aureus* (MRSA) is much more difficult to treat and comprises 20% to 74% of the SA infections around the world (7). Of great concern is that 6% to 27% of hospital-treated skin infections lead to bloodstream infections (BSIs) and sepsis (5), MRSA infections having a 50% greater chance than methicillin-sensitive *S. aureus* (MSSA) infections for this outcome, with 15% to 60% mortality (8). Beyond ABSSSIs and BSIs, there are growing medical needs to treat other hospital infections caused by multidrug-resistant *S. pyogenes*, especially *Enterococcus faecalis* and *faecium*, where mortality is between 20% and 50% (9, 10).

While serious hospital infections are almost always treated with an IV drug immediately, switching to an oral step-down therapy after initial success with IV administration has major medical benefits and cost savings, and it is ideal when the switch is to the same agent (11, 12). The longstanding agent of choice for treating serious hospitalized gram-positive infections is vancomycin (no oral step down); however, rates of resistance or intermediate resistance to SA strains have risen to >5% and continue to climb (9, 13). While new antibiotics covering gram-positive infections have been introduced in the past decade, all but one are from established drug classes (fluoroquinolones, tetracyclines, oxazolidinones, and glycopeptides), and the risk of resistance remains significant for these classes (14, 15). As of 2021, there were 36 drugs approved for ABSSSIs (14, 15), with 32 new compounds in the pipeline, but only three of the pipeline compounds are small molecules with new mechanisms of action (16). Despite all the approved agents for ABSSSIs, its management remains challenging with inappropriate treatment in up to 25% of cases (17). Many of the treatments are only moderately effective, due in large part to the emergence of resistant bacteria and very few new compounds with novel mechanisms of action (2, 14, 17).

We recently reported a new, broad-spectrum gram-positive agent, MRS-2541, for oral and IV administration (18). MRS-2541 is a highly potent (sub-nanomolar) small molecule inhibitor of type 1 methionyl-tRNA synthetase, which is an essential enzyme conserved across gram-positive pathogens, including MRSA and multidrug-resistant enterococci. The type 2 methionyl-tRNA synthetase is present in gram-negative bacteria and the cytosol of eukaryotes. A few gram-positive species, like *S. pneumoniae*, have both enzymes (19). As expected, MRS-2541 is inactive vs gram-negative bacteria and less active vs organisms that contain both enzymes (18). The MRS enzymes attach methionine to the corresponding tRNA$^{met}$ that is used at the ribosome for protein initiation and translation. Against an initial test panel, MRS-2541 displayed MICs versus *S. aureus* and *E. faecalis* of ≤0.125 µg/mL with a >100-fold selectivity index in mammalian cell cytotoxicity assays. It had good oral bioavailability with extensive tissue distribution in rodents and showed efficacy comparable to linezolid, with both oral and subcutaneous dosing in murine models (18). While the initial MICs vs laboratory strains indicated the potential for broad gram-positive activity, we desired to examine a larger panel of clinical isolates biased with resistant phenotypes, which, though rare in the general population, provide valuable information on the full spectrum of MRS-2541. Most of the previous MetRS inhibitor literature (18, 20, 21) accentuated the excellent *S. aureus* and MRSA activity, but for the first time, we also report on a full panel of staphylococci species as well as MDR streptococci and enterococci, given their growing role in gram-positive hospital infections (9, 10).

## RESULTS

The test article MRS-2541 was evaluated for *in vitro* activity alongside the appropriate comparators against large panels of gram-positive isolates, including those with defined resistance phenotypes. The test panels were enriched for antimicrobial resistance (AMR) phenotypes (i.e., MRSA, vancomycin-intermediate *S. aureus* [VISA], vancomycin-resistant *Enterococcus* [VRE], and penicillin-resistant *S. pneumoniae* [PRSP]) that can be seen in the line listings, along with individual MIC values and quality control (QC) testing (Tables S1 to S3). The results of testing MRS-2541 can be found in Tables 1 and 2 (staphylococci), Table 3 (enterococci), and Tables 4 and 5 (streptococci).

### Staphylococci

As shown in Table 1, MRS-2541 demonstrated an $MIC_{50/90}$ of 0.03/0.06 µg/mL overall against the 45 *S. aureus* isolates tested. This same $MIC_{50/90}$ was observed against the methicillin-susceptible subset (MSSA, $N = 18$), the methicillin-resistant (MRSA; $N = 27$), and vancomycin-intermediate/heterogeneous vancomycin-intermediate (VISA/hVISA; $N = 12$) subsets of isolates. The MRS-2541 MIC values were lower overall than those observed with the seven comparator antibiotics against the 45 *S. aureus* isolates, as well as the MRSA and MSSA subsets. Levofloxacin, oxacillin, erythromycin, clindamycin, vancomycin, daptomycin, and linezolid had the expected activity, with $MIC_{90}$ values of >16, >32, >16, >16, 4, 2, and 2 µg/mL, respectively, against the 45 *S. aureus* isolates. More than 50% of the isolates in this set of *S. aureus* were resistant to levofloxacin; 60% were resistant to oxacillin, and 66.7% were erythromycin resistant.

MRS-2541 had the most potent activity against the 35 coagulase-negative *S. aureus* (CoNS) when examined against the seven comparator compounds, with $MIC_{50}$ and $MIC_{90}$ values of 0.12 µg/mL (Table 1). These same $MIC_{50/90}$ values were observed for MRS-2541 against the 14 methicillin-susceptible CoNS and the 21 methicillin-resistant CoNS. Sixty

**TABLE 1** Summary of the activity of MRS-2541 and comparators against *Staphylococcus* spp. isolates[a]

| Organism | N | Drug | Phenotype | MIC (µg/mL) | | | S/I/R (%) |
|---|---|---|---|---|---|---|---|
| | | | | Range | $MIC_{50}$ | $MIC_{90}$ | |
| *S. aureus* | 45 | MRS-2541 | Overall | ≤0.016 to 0.12 | 0.03 | 0.06 | – |
| | | | MSSA ($N = 18$) | ≤0.016 to 0.12 | 0.03 | 0.06 | – |
| | | | MRSA ($N = 27$) | ≤0.016 to 0.06 | 0.03 | 0.06 | – |
| | | | VISA/hVISA ($N = 12$) | ≤0.016 to 0.06 | 0.03 | 0.06 | – |
| | | Levofloxacin | Overall | 0.03 to >16 | 2 | >16 | 44.4/4.4/51.1 |
| | | Oxacillin | Overall | 0.12 to >32 | 16 | >32 | 40.0/–/60.0 |
| | | Erythromycin | Overall | 0.25 to >16 | >16 | >16 | 33.3/0/66.7 |
| | | Clindamycin | Overall | 0.06 to >16 | 0.12 | >16 | 66.7/0/33.3 |
| | | Vancomycin | Overall | 0.25 to 8 | 0.5 | 4 | 77.8/22.2/0.0 |
| | | Daptomycin | Overall | 0.5 to 4 | 1 | 2 | 86.7/–/– |
| | | Linezolid | Overall | 1 to 4 | 2 | 2 | 100/–/0.0 |
| CoNS | 35 | MRS-2541 | Overall | 0.06 to 0.25 | 0.12 | 0.12 | – |
| | | | MSCoNS ($N = 14$) | 0.06 to 0.12 | 0.12 | 0.12 | – |
| | | | MRCoNS ($N = 21$) | 0.06 to 0.25 | 0.12 | 0.12 | – |
| | | Levofloxacin | Overall | 0.06 to >16 | 2 | 16 | 48.6/5.7/45.7 |
| | | Oxacillin | Overall | 0.06 to >32 | 4 | >32 | 40.0/–/60.0 |
| | | Erythromycin | Overall | 0.12 to >16 | >16 | >16 | 2.9/22.8/74.3 |
| | | Clindamycin | Overall | 0.06 to >16 | 0.12 | >16 | 62.9/2.9/34.3 |
| | | Vancomycin | Overall | 0.5 to 2 | 1 | 2 | 100/0.0/0.0 |
| | | Daptomycin | Overall | 0.25 to 1 | 0.5 | 1 | 100/–/– |
| | | Linezolid | Overall | 0.5 to 2 | 1 | 1 | 100/–/0.0 |

[a]CoNS, coagulase-negative *S. aureus*; I, intermediate; MRSA, methicillin-resistant *Staphylococcus aureus*; MRCoNS, methicillin-resistant CoNS; MSCoNS, methicillin-sensitive CoNS; MSSA, methicillin-sensitive *S. aureus*; VISA, vancomycin-intermediate *S. aureus*; R, resistant; S, susceptible; –, no interpretive criteria per Clinical and Laboratory Standards Institute.

**TABLE 2** Summary of the activity of MRS-2541 and comparators against additional *Staphylococcus* spp. isolates[a]

| Organism | N | Drug | MIC (µg/mL) | | | S/I/R (%) |
| --- | --- | --- | --- | --- | --- | --- |
| | | | Range | $MIC_{50}$ | $MIC_{90}$ | |
| *Staphylococcus epidermidis* | 15 | MRS-2541 | 0.06 to 0.12 | 0.06 | 0.12 | – |
| | | Levofloxacin | 0.12 to >16 | 8 | >16 | 33.3/6.7/60.0 |
| | | Oxacillin | 0.06 to >32 | 4 | 16 | 26.7/–/73.3 |
| | | Erythromycin | 0.25 to >16 | >16 | >16 | 0/26.7/73.3 |
| | | Clindamycin | 0.12 to >16 | 2 | >16 | 46.7/6.7/46.7 |
| | | Vancomycin | 1 to 2 | 1 | 2 | 100/0/0 |
| | | Daptomycin | 0.5 to 1 | 1 | 1 | 100/–/– |
| | | Linezolid | 0.5 to 1 | 1 | 1 | 100/–/0 |
| *Staphylococcus haemolyticus* | 10 | MRS-2541 | 0.06 to 0.12 | 0.12 | 0.12 | – |
| | | Levofloxacin | 0.06 to 16 | 0.12 | 8 | 70.0/0.0/30.0 |
| | | Oxacillin | 0.12 to >32 | 1 | >32 | 40.0/–/60.0 |
| | | Erythromycin | 0.12 to >16 | >16 | >16 | 10.0/10.0/80.0 |
| | | Clindamycin | 0.06 to 4 | 0.06 | 0.25 | 90.0/0.0/10.0 |
| | | Vancomycin | 0.5 to 2 | 0.5 | 1 | 100/0.0/0.0 |
| | | Daptomycin | 0.25 to 0.5 | 0.25 | 0.5 | 100/–/– |
| | | Linezolid | 0.5 to 1 | 0.5 | 1 | 100/–/0.0 |
| *Staphylococcus hominis* | 10 | MRS-2541 | 0.06 to 0.25 | 0.12 | 0.25 | – |
| | | Levofloxacin | 0.06 to 8 | 0.25 | 8 | 50.0/10.0/40.0 |
| | | Oxacillin | ≤0.03 to >32 | 0.25 | >32 | 60.0/0.0/40.0 |
| | | Erythromycin | 0.12 to >16 | >16 | >16 | 30.0/0.0/70.0 |
| | | Clindamycin | 0.06 to >16 | 0.012 | >16 | 60.0/0.0/40.0 |
| | | Vancomycin | 0.25 to 1 | 0.5 | 1 | 100/0.0/0.0 |
| | | Daptomycin | 0.25 to 0.5 | 0.25 | 0.5 | 100/–/– |
| | | Linezolid | 0.5 to 2 | 1 | 2 | 100/–/0.0 |

[a]I, intermediate; R, resistant; S, susceptible; –, no interpretive criteria per Clinical and Laboratory Standards Institute.

percent of these isolates were oxacillin resistant; 45.7% were resistant to levofloxacin; and 74.3% were resistant to erythromycin. The comparators oxacillin, levofloxacin, and erythromycin had appropriately high $MIC_{90}$ values of >32, 16, and >16 µg/mL, respectively. Among these isolates, there was no resistance to vancomycin, daptomycin, or linezolid, and these $MIC_{90}$ values were 1 to 2 µg/mL.

Table 2 summarizes the activity of MRS-2541 against isolates of *Staphylococcus epidermidis* (N = 15), *Staphylococcus haemolyticus* (N = 10), and *Staphylococcus hominis* (N = 10). MRS-2541 exhibited $MIC_{50/90}$ values of 0.06/0.12 µg/mL against the *S. epidermidis* tested overall; this compares with $MIC_{90}$ values of 16 to >16 µg/mL for levofloxacin, oxacillin, erythromycin, and clindamycin (resistance rates of 60.0, 73.3, 73.3, and 46.7%, respectively), and 1 to 2 µg/mL for the remaining comparators, against which there was no resistance in this set of isolates. Against the 10 tested *S. haemolyticus* and the 10 tested *S. hominis*, MRS-2541 had $MIC_{50/90}$ values of 0.12/0.12 and 0.12/0.25 µg/mL, respectively. Both sets of isolates had elevated levels of resistance to oxacillin and erythromycin, and these drugs had associated elevated MIC values. Comparators for which there was no resistance among these isolates (vancomycin, daptomycin, or linezolid) showed $MIC_{90}$ values of 0.5 to 2 µg/mL.

## Enterococci

MRS-2541 against 15 *E. faecalis* demonstrated potent activity against these isolates overall, with $MIC_{50}$ and $MIC_{90}$ values of ≤0.016 µg/mL, including against four VRE isolates (Table 3). Resistance to erythromycin in this group was 46.7%; levofloxacin resistance was 53.3%; and vancomycin resistance was 26.7%. These comparators exhibited $MIC_{90}$

**TABLE 3** Summary of activity of MRS-2541 and comparators against *Enterococcus* spp. isolates[a]

| Organism | N | Drug | Phenotype | MIC (µg/mL) | | | S/I/R (%) |
|---|---|---|---|---|---|---|---|
| | | | | Range | MIC$_{50}$ | MIC$_{90}$ | |
| *E. faecalis* | 15 | MRS-2541 | Overall | ≤0.016 to ≤0.016 | ≤0.016 | ≤0.016 | – |
| | | | VSE (N = 11) | ≤0.016 to ≤0.016 | ≤0.016 | ≤0.016 | – |
| | | | VRE (N = 4) | ≤0.016 to ≤0.016 | ≤0.016 | – | – |
| | | Levofloxacin | Overall | 0.5 to >16 | 8 | >16 | 40.0/6.7/53.3 |
| | | Ampicillin | Overall | 1 to 2 | 1 | 2 | 100/–/0.0 |
| | | Erythromycin | Overall | 0.12 to >16 | 4 | >16 | 20.0/33.3/46.7 |
| | | Clindamycin | Overall | 8 to >16 | >16 | >16 | – |
| | | Vancomycin | Overall | 0.5 to >32 | 1 | >32 | 73.3/0.0/26.7 |
| | | Daptomycin | Overall | 0.5 to 2 | 1 | 2 | 100/0.0/0.0 |
| | | Linezolid | Overall | 1 to 2 | 2 | 2 | 100/0.0/0.0 |
| *E. faecium* | 15 | MRS-2541 | Overall | ≤0.016 to >16 | ≤0.016 | >16 | – |
| | | | VSE (N = 10) | ≤0.016 to >16 | ≤0.016 | >16 | – |
| | | | VRE (N = 5) | ≤0.016 to ≤0.016 | ≤0.016 | – | – |
| | | Levofloxacin | Overall | 0.03 to >16 | >16 | >16 | 33.3/0.0/66.7 |
| | | Ampicillin | Overall | 1 to >32 | >32 | >32 | 40.0/–/60.0 |
| | | Erythromycin | Overall | 1 to >16 | >16 | >16 | 0.0/20.0/80.0 |
| | | Clindamycin | Overall | 0.06 to >16 | >16 | >16 | – |
| | | Vancomycin | Overall | 0.5 to >32 | 0.5 | >32 | 60.0/6.7/33.3 |
| | | Daptomycin | Overall | 0.12 to 8 | 2 | 4 | –/93.3*/6.7 |
| | | Linezolid | Overall | 2 to 8 | 4 | 4 | 33.3/60.0/6.7 |

[a]I, intermediate; R, resistant; S, susceptible; VRE, vancomycin-resistant *Enterococcus*; VSE, vancomycin-susceptible *Enterococcus*; –, no interpretive criteria; *, susceptible dose dependent per Clinical and Laboratory Standards Institute.

values of >16, >16, and >32 µg/mL, respectively. The MIC$_{50/90}$ values for daptomycin and linezolid were 1/2 and 2/2 µg/mL, respectively.

Against 15 *E. faecium* isolates, MRS-2541 had an MIC$_{50/90}$ value of ≤0.016/>16 µg/mL. Against the 5 VRE isolates, the test article displayed MIC values of ≤0.016 µg/mL, whereas against the 10 VSE isolates, the MIC$_{90}$ value for MRS-2541 was >16 µg/mL due to MIC values of >16 µg/mL against 2 of the VSEs tested. Sixty percent of these isolates were ampicillin resistant; 66.7% were resistant to levofloxacin; and 80% were resistant to erythromycin. The comparators ampicillin, levofloxacin, and erythromycin had elevated MIC$_{90}$ values of >32, >16, and >16 µg/mL, respectively. Daptomycin and linezolid had MIC$_{50/90}$ values of 2/4 and 4/4 µg/mL, respectively.

## Streptococci

Despite the fact that some *S. pneumoniae* strains possess the type 2 MetRS enzyme, which is not sensitive to any of MetRS inhibitors, MRS-2541 had MIC$_{50/90}$ values of 0.25/0.5 µg/mL against the 30 *S. pneumoniae* isolates tested, the same value as those observed with vancomycin (0.25/0.5 µg/mL) against this panel (Table 4). This MIC$_{50/90}$ value was the same against the 12 penicillin-susceptible *S. pneumoniae* and similar to that against the 13 multidrug-resistant/penicillin-resistant *S. pneumoniae* (MDR/PRSP, 0.5/0.5 µg/mL). The MIC$_{50}$ for MRS-2541 against the five penicillin-intermediate *S. pneumoniae* (PISP) isolates tested was 0.5 µg/mL. With only low or no levels of resistance in this panel, levofloxacin, vancomycin, daptomycin, and linezolid displayed good activity (MIC$_{90}$ values 0.12 to 1 µg/mL); resistance levels to penicillin, erythromycin, and clindamycin were 43.3% to 50%, resulting in MIC$_{90}$ values of 4 to >16 µg/mL for these drugs.

MRS-2541 had MIC$_{50/90}$ values of 2/8 µg/mL against the 30 beta-hemolytic *Streptococcus* isolates overall. There was a 43.3% resistance to erythromycin among the isolates in this set. Against the 13 erythromycin-resistant isolates in this panel, activity for MRS-2541 was 0.5 to 8 µg/mL with an MIC$_{90}$ of 8 µg/mL. The same MIC$_{90}$ value of 8

**TABLE 4** Summary of activity of MRS-2541 and comparators against *Streptococcus* spp. isolates[a]

| Organism | N | Drug | Phenotype | MIC (µg/mL) | | | S/I/R (%) |
|---|---|---|---|---|---|---|---|
| | | | | Range | MIC$_{50}$ | MIC$_{90}$ | |
| *S. pneumoniae* | 30 | MRS-2541 | Overall | 0.06 to >16 | 0.25 | 0.5 | – |
| | | | PSSP (*N* = 12) | 0.06 to 0.5 | 0.25 | 0.5 | – |
| | | | PISP (*N* = 5) | 0.25 to >16 | 0.5 | – | – |
| | | | PRSP/MDRSP (*N* = 13) | 0.25 to >16 | 0.5 | 0.5 | – |
| | | Levofloxacin | Overall | 0.5 to 8 | 1 | 1 | 96.7/0.0/3.3 |
| | | Penicillin | Overall | ≤0.004 to >4 | 0.25 | 4 | 40.0/16.7/43.3 |
| | | Erythromycin | Overall | ≤0.016 to >16 | 0.25 | >16 | 50.0/0.0/50.0 |
| | | Clindamycin | Overall | ≤0.016 to >16 | 0.06 | >16 | 53.3/0.0/46.7 |
| | | Vancomycin | Overall | ≤0.03 to 0.5 | 0.25 | 0.5 | 100/–/– |
| | | Daptomycin | Overall | 0.06 to 0.25 | 0.12 | 0.12 | |
| | | Linezolid | Overall | 0.12 to 1 | 0.5 | 1 | 100/–/– |
| Beta-hemolytic streptococci | 30 | MRS-2541 | Overall | 0.5 to 8 | 2 | 8 | – |
| | | | ERY-S (*N* = 15) | 0.5 to 8 | 0.5 | 8 | – |
| | | | ERY-R (*N* = 13) | 0.5 to 8 | 8 | 8 | – |
| | | Levofloxacin | Overall | 0.25 to 2 | 0.5 | 1 | 100/0.0/0.0 |
| | | Penicillin | Overall | ≤0.004 to 2 | 0.016 | 0.03 | 96.6/–/– |
| | | Erythromycin | Overall | ≤0.016 to >16 | 0.06 | >16 | 50.0/6.7/43.3 |
| | | Clindamycin | Overall | 0.03 to >16 | 0.06 | >16 | 76.7/0.0/23.3 |
| | | Vancomycin | Overall | 0.25 to 2 | 0.25 | 0.5 | 96.6/–/– |
| | | Daptomycin | Overall | 0.03 to 1 | 0.12 | 0.25 | 100/–/– |
| | | Linezolid | Overall | 0.5 to 1 | 1 | 1 | 100/–/– |
| Viridans group streptococci | 30 | MRS-2541 | Overall | ≤0.016 to >16 | 0.25 | 16 | – |
| | | Levofloxacin | Overall | 0.5 to 16 | 1 | 2 | 93.3/0.0/6.7 |
| | | Penicillin | Overall | ≤0.004 to 4 | 0.06 | 2 | 63.3/33.3/3.3 |
| | | Erythromycin | Overall | ≤0.016 to >16 | 0.03 | 4 | 63.3/10.0/26.7 |
| | | Clindamycin | Overall | ≤0.016 to >16 | 0.03 | 16 | 73.3/0.0/26.7 |
| | | Vancomycin | Overall | 0.25 to 8 | 0.5 | 1 | 93.3/–/– |
| | | Daptomycin | Overall | 0.03 to 2 | 0.25 | 2 | 86.7/–/– |
| | | Linezolid | Overall | 0.25 to 2 | 1 | 2 | 100/–/– |

[a]I, intermediate; MDRSP, multidrug-resistant *S. pneumoniae*; PISP, penicillin-intermediate *S. pneumoniae*; PRSP, penicillin-resistant *S. pneumoniae*; PSSP, penicillin-susceptible *S. pneumoniae*; R, resistant; S, susceptible; –, no interpretive criteria per CLSI.

µg/mL was observed for this test article against the 15 erythromycin-susceptible isolates in this set. Resistance rates to other drugs were low; MIC$_{90}$ values for levofloxacin, penicillin, vancomycin, daptomycin, and linezolid were 0.03 to 1 µg/mL. If individual beta-hemolytic species are considered, MRS-2541 had strong activity when testing against *S. pyogenes* (MIC$_{50/90}$, 0.5/0.5 µg/mL) and relatively decreased activity against *S. agalactiae* (MIC$_{50/90}$, 8/8 µg/mL) (Table 5).

Against viridans group streptococci, the MIC$_{50}$ and MIC$_{90}$ for MRS-2541 were 0.25 and 16 µg/mL, respectively. About one-quarter of the isolates were resistant to erythromycin or clindamycin; rates of resistance to other agents were low. Levofloxacin, penicillin, vancomycin, daptomycin, and linezolid had MIC$_{90}$ values of 1 to 2 µg/mL, and those for erythromycin and clindamycin were 4 and 16 µg/mL, respectively. Elevated MIC values were observed in *S. mitis* (MIC$_{50/90}$, 0.5/>16 µg/mL) but not in *S. salivarius* (MIC range of 0.06 to 2 µg/mL) or *S. sanguinis* (MIC range of ≤0.016 to 1) (Table 5).

## DISCUSSION

New therapies with novel targets are desperately needed to treat antibiotic-resistant bacteria. A recent study estimated that 4.71 million deaths were associated with bacterial AMR in 2019 (22). This same group in 2024 estimated that 92 million deaths could be averted before 2050 through better treatment of infections and improved access to

**TABLE 5** Summary of activity of MRS-2541 and comparators against additional *Streptococcus* spp. isolates[a]

| Organism | N | Drug | MIC (µg/mL) | | | S/I/R (%) |
|---|---|---|---|---|---|---|
| | | | Range | $MIC_{50}$ | $MIC_{90}$ | |
| *S. pyogenes* | 15 | MRS-2541 | 0.5 to 2 | 0.5 | 0.5 | – |
| | | Levofloxacin | 0.5 to 2 | 0.5 | 1 | 100/0.0/0.0 |
| | | Penicillin | ≤0.004 to 0.016 | 0.008 | 0.016 | 100/–/– |
| | | Erythromycin | ≤0.016 to >16 | 0.03 | >16 | 80.0/0.0/20.0 |
| | | Clindamycin | 0.03 to >16 | 0.06 | >16 | 80.0/0.0/20.0 |
| | | Vancomycin | 0.25 to 0.5 | 0.25 | 0.5 | 100/–/– |
| | | Daptomycin | 0.03 to 0.25 | 0.03 | 0.12 | 100/–/– |
| | | Linezolid | 0.5 to 1 | 1 | 1 | 100/–/– |
| *S. agalactiae* | 15 | MRS-2541 | 8 to 8 | 8 | 8 | – |
| | | Levofloxacin | 0.25 to 1 | 0.5 | 1 | 100/0.0/0.0 |
| | | Penicillin | 0.016 to 2 | 0.03 | 0.03 | 93.3/–/– |
| | | Erythromycin | 0.03 to >16 | 2 | >16 | 20.0/13.3/66.7 |
| | | Clindamycin | 0.03 to >16 | 0.06 | >16 | 73.3/0.0/26.7 |
| | | Vancomycin | 0.25 to 2 | 0.5 | 0.5 | 93.3/–/– |
| | | Daptomycin | 0.12 to 1 | 0.25 | 0.25 | 100/–/– |
| | | Linezolid | 0.5 to 1 | 1 | 1 | 100/–/– |
| *S. mitis* | 13 | MRS-2541 | 0.12 to >16 | 0.5 | >16 | – |
| | | Levofloxacin | 1 to 8 | 1 | 1 | 92.3/0.0/7.7 |
| | | Penicillin | 0.0008 to 4 | 0.12 | 2 | 61.5/30.8/7.7 |
| | | Erythromycin | ≤0.016 to >16 | 0.5 | >16 | 46.2/7.7/46.2 |
| | | Clindamycin | ≤0.016 to >16 | 0.03 | 16 | 84.6/0.0/15.4 |
| | | Vancomycin | 0.25 to 0.5 | 0.5 | 0.5 | 100/–/– |
| | | Daptomycin | 0.25 to 1 | 0.25 | 1 | 100/–/– |
| | | Linezolid | 0.25 to 1 | 1 | 1 | 100/–/– |
| *S. salivarius* | 6 | MRS-2541 | 0.06 to 2 | 0.12 | – | – |
| | | Levofloxacin | 0.5 to 2 | 1 | – | 100/0.0/0.0 |
| | | Penicillin | ≤0.004 to 0.25 | 0.06 | – | 83.3/16.7/0 |
| | | Erythromycin | ≤0.016 to 0.03 | ≤0.016 | – | 100/0.0/0.0 |
| | | Clindamycin | ≤0.016 to 0.03 | ≤0.016 | – | 100/0.0/0.0 |
| | | Vancomycin | 0.5 to 1 | 1 | – | 100/–/– |
| | | Daptomycin | 0.03 to 0.5 | 0.12 | – | 100/–/– |
| | | Linezolid | 0.5 to 2 | 2 | – | 100/–/– |
| *S. sanguinis* | 11 | MRS-2541 | ≤0.016 to 1 | 0.25 | 0.5 | – |
| | | Levofloxacin | 0.5 to 16 | 1 | 1 | 90.9/0.0/9.1 |
| | | Penicillin | 0.03 to 2 | 0.12 | 2 | 54.5/45.5/0.0 |
| | | Erythromycin | ≤0.016 to >16 | 0.25 | 2 | 63.6/18.2/18.2 |
| | | Clindamycin | ≤0.016 to >16 | 2 | >16 | 45.5/0/54.5 |
| | | Vancomycin | 0.25 to 8 | 0.5 | 8 | 81.8/–/– |
| | | Daptomycin | 0.12 to 8 | 0.25 | 8 | 63.6/–/– |
| | | Linezolid | 0.5 to 2 | 1 | 2 | 100/–/– |

[a]I, intermediate; R, resistant; S, susceptible; –, no interpretive criteria, per CLSI.

antibiotics (22). The Centers for Disease Control and Prevention (CDC) estimated that annual costs due to AMR can be upwards of $4.6 billion (23).

Whereas vancomycin is often prescribed for serious gram-positive infections, including those caused by staphylococci, enterococci, and streptococci, resistance rates are increasing for this antibiotic. Both gram-negative and gram-positive infections require new therapy options to close this gap. *Staphylococcus aureus* and enterococci are the second and fifth most common pathogens causing healthcare-associated infections (HAI). VRE infections account for greater than 50,000 HAIs, and approximately 80% of *E. faecium* HAIs are vancomycin resistant (24–26). With *E. faecium* infections exhibiting high prevalence of vancomycin and β-lactam resistance, additional therapeutic options are

needed, especially since linezolid resistance has also been observed to be increasing in enterococci (27).

MRS-2541 is currently under preclinical development for gram-positive ABSSSIs and BSIs. This compound was evaluated for *in vitro* activity alongside appropriate comparators against large panels of gram-positive isolates, enriched for isolates with antibiotic resistance phenotypes. Against isolates of clinical relevance of *S. aureus* (including MRSA and VISA) and coagulase-negative *Staphylococcus*, MRS-2541 was the most potent article evaluated. MRS-2541 also demonstrated the most potent $MIC_{50/90}$ values against the panels of *E. faecalis* and *E. faecium* tested, including vancomycin-resistant isolates; there were only two *E. faecium* strains where MRS-2541 was not active (MIC >16 µg/mL). This is the first report of enterococcal strains that were not sensitive to a MetRS inhibitor. *E. faecium* is known to have a multitude of resistance mechanisms, which could include a MetRS enzyme with a modified active site or more likely upregulated efflux pumps, which is often the reason for lack of sensitivity to a new agent (28, 29). Despite the presence of the MetRS type 2 enzyme in some *S. pneumoniae* (19), MRS-2541 showed good activity overall and behaved similarly against PISP and MDR/PRSP but displayed weaker activity against *S. agalactiae and S. mitis*. All 15 strains of *S. agalactiae* had MICs of 8 µg/mL. This could imply sequence modifications of the MetRS enzyme, efflux pumps that recognize MRS-2541, species differences in gene regulation (30), or a combination of factors. Our inspection of all *S. agalactiae* genomes curated in UniProt ([www.uniprot.org](http://www.uniprot.org)) does not indicate the presence of a type 2 MetRS enzyme, although the sampling is limited to nine strains. For *S. mitis*, only 4 of 13 strains were insensitive. Further study is needed to understand the mechanism(s) of the resistance, although, as with *S. agalactiae*, there is no evidence of the presence of a type 2 MetRS based on six *S. mitis* genomes in Uniprot. It should be noted that all previously reported MetRS inhibitors are significantly less active against streptococci (16- to 64-fold) relative to their potency versus staphylococci (18, 20, 21). The good activity against *S. pyogenes* ($MIC_{90}$ 0.5 µg/mL) is promising for the desired treatment of ABSSSIs, but for BSIs, identifying the causative organism may be important to ensure successful treatment.

In summary, MRS-2541 demonstrated potent activity against a challenge panel of drug-resistant staphylococci, enterococci, and streptococci, including the lowest $MIC_{50/90}$ values for *S. aureus*, CoNS, and enterococci compared to other standard-of-care agents. Potency did not appear to be affected by other resistance phenotypes, including MRSA, VISA, VRE, and PRSP, showing promise for this agent in the context of antibiotic-resistant infections. These data show the promise of MRS-2541 as a novel compound in an environment of increasing resistance rates and fewer therapeutic options for severe gram-positive infections and position this agent well for its intended indications.

## MATERIALS AND METHODS

### Test organisms

Test organisms consisted of reference strains from the American Type Culture Collection (ATCC, Manassas, VA), the CDC (Atlanta, GA), the Network on Antimicrobial Resistance in *Staphylococcus aureus* (NARSA; BEI Resources, managed by ATCC), and independent non-consecutive clinical isolates from the Microbiologics Repository (MMX, Kalamazoo, MI). The CDC, NARSA, and a subset of the MMX isolates were included to select for particular underrepresented susceptibility phenotypes. QC isolates, including *S. aureus* ATCC 29,213, *E. faecalis* ATCC 29,212, and *S. pneumoniae* ATCC 49,619, were included as necessary during testing (31, 32). MIC values were determined in singleton.

### Test media

Cation-adjusted Mueller-Hinton Broth (CAMHB) was used for MIC testing as described by Clinical and Laboratory Standards Institute (CLSI) (31, 32). CAMHB was prepared by supplementing Mueller-Hinton broth (Becton Dickinson) to a final concentration of 20

mg/L Ca$^{2+}$ (Sigma) and 12.5 mg/L Mg$^{2+}$ (EMD Millipore). For the testing of *Streptococcus* spp., this medium was supplemented with 3% laked horse blood (Hemostat).

## Broth microdilution MIC assay

The MIC assay method followed the procedure described by CLSI (31, 32) and employed automated liquid handlers to conduct serial dilutions and liquid transfers. Automated liquid handlers included the Multidrop 384 (Labsystems, Helsinki, Finland), Biomek 3000, and Biomek FX (Beckman Coulter, Fullerton, CA).

The stock solutions for the comparators were prepared using solvents and diluents recommended by CLSI (32). MRS-2541 was supplied by TSRL and dissolved in DMSO for testing. Levofloxacin, oxacillin, ampicillin, penicillin, erythromycin, clindamycin, and vancomycin were supplied by Sigma. Daptomycin and linezolid were purchased from Selleck Chemical (Houston, TX).

Compounds were evaluated with twofold dilutions across an 11-pt scale. The wells of column 12 contained no drug and served as the organism growth control wells for each respective diluent. Each test plate contained the respective media and test agents for broth microdilution evaluation.

A standardized inoculum of each organism was prepared per CLSI methods (31, 32). Colonies were picked from the primary plate, and a suspension was prepared to equal a 0.5 McFarland turbidity standard. The Biomek 3000 delivered diluted standardized inoculum into each well, targeting a final concentration of ~5 × 10$^5$ CFU/mL. Plates inoculated with bacteria were incubated for approximately 20 hours at 35℃. The MIC was visually read and recorded as the lowest concentration where bacterial growth was inhibited.

## ACKNOWLEDGMENTS

The authors gratefully acknowledge funding support from NIH R44AI134190 to N.M.R.M., F.S.B., Z.Z., E.F., E.L., D.R., and J.D.; and R01AI152358 to N.M.R.M., F.S.B., Z.Z., and E.F.

We appreciate contributions to the research from J. Robert Gillespie, Aisha Mushtaq, and Sayaka Shibata.

## AUTHOR AFFILIATIONS

[1]TSRL Inc., Ann Arbor, Michigan, USA
[2]Microbiologics, Inc., Bacteriology and Mycology Center of Excellence, Portage, Michigan, USA
[3]CERID, Division of Allergy and Infectious Diseases, University of Washington, Seattle, Washington, USA
[4]Department of Biochemistry, University of Washington, Seattle, Washington, USA

## AUTHOR ORCIDs

David A. Hufnagel  http://orcid.org/0009-0003-4878-1520
Frederick S. Buckner  http://orcid.org/0000-0001-7796-6477

## FUNDING

| Funder | Grant(s) | Author(s) |
| --- | --- | --- |
| National Institutes of Health | R01 AI152358 | Erkang Fan |
| | | Frederick S. Buckner |
| National Institutes of Health | R44 AI134190 | Elke Lipka |

## AUTHOR CONTRIBUTIONS

John Domagala, Conceptualization, Formal analysis, Project administration, Supervision, Writing – original draft, Writing – review and editing | Seyedhameneh Jahanbakhsh, Investigation, Methodology | Angela K. Mendez, Investigation | Chris M. Pillar, Investigation | Andrea Marra, Investigation | David A. Hufnagel, Formal analysis, Funding acquisition, Investigation, Methodology, Project administration, Resources, Writing – review and editing | Nora M. R. Molasky, Investigation, Writing – review and editing | Zhongsheng Zhang, Investigation, Writing – review and editing | Erkang Fan, Conceptualization, Funding acquisition, Project administration, Resources, Writing – review and editing | Dawn Reyna, Investigation, Project administration, Writing – review and editing | Elke Lipka, Conceptualization, Funding acquisition, Project administration, Supervision, Writing – review and editing | Frederick S. Buckner, Conceptualization, Funding acquisition, Project administration, Writing – review and editing

## DATA AVAILABILITY

All data are readily available by contacting the corresponding author.

## ADDITIONAL FILES

The following material is available online.

### Supplemental Material

**Supplemental tables (Spectrum03615-25-s0001.xlsx).** Tables S1 to S3.

### Open Peer Review

**PEER REVIEW HISTORY (review-history.pdf).** An accounting of the reviewer comments and feedback.

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
