## [Reviewer comments · Microbiology Spectrum]

Microbiology Spectrum

In vitro activity of MRS-2541, a novel MetRS inhibitor, against a selection of resistant Gram-positive organisms associated with serious hospital infections

John Domagala, Seyedehameneh Jahanbakhsh, Angela Kennedy-Mendez, Chris Pillar, Andrea Marra, David Hufnagel, Nora Molasky, Zhongsheng Zhang, Erkang Fan, Dawn Reyna, Elke Lipka, and Frederick Buckner

Corresponding Author(s): Frederick Buckner, University of Washington

Review Timeline:

Submission Date:	November 11, 2025
Editorial Decision:	January 5, 2026
Revision Received:	February 14, 2026
Accepted:	February 17, 2026

Editor: Mark Pandori

Reviewer(s): The reviewers have opted to remain anonymous.

Transaction Report:

DOI: <https://doi.org/10.1128/spectrum.03615-25>

Re: Spectrum03615-25 (In vitro activity of MRS-2541, a novel MetRS inhibitor, against a selection of resistant Gram-positive organisms associated with serious hospital infections)

Dear Prof. Frederick Buckner:

Thank you for the privilege of reviewing your work. Below you will find my comments, instructions from the Spectrum editorial office, and the reviewer comments.

Note: You will find Reviews from Reviewers #2 and #3 only. Reviewer #1 did not complete review of the manuscript on time and asked to be relieved of the task upon reaching the deadline. An additional Reviewer (#3) was identified and accepted the task of review. Thank you and sorry for the delay.

Revision Guidelines

Sincerely,
Mark Pandori
Editor
Microbiology Spectrum

Reviewer #2 (Comments for the Author):

This manuscript by Domagala and co-workers describes the results of studies to evaluate the in vitro antibacterial activity of the

novel MetRS inhibitor, MRS-2541, against extended panels of clinically important Gram-positive bacterial pathogens. The data indicate that MRS-2541 generally has potent antibacterial activity (low MICs) against isolates of *Staphylococcus* spp., *Enterococcus* spp. and *Streptococcus* spp., including drug-resistant strains.

I have the following comments and suggestions for the authors:

1. Figure 1 is unnecessary because the chemical structure of MRS-2541 has previously been published.
2. MIC data for the QC strains should be included as a separate line listing in the tables so that readers are assured the antibacterial susceptibility assay is performing within the expected (CLSI) guidelines. Also, the specific identities of the QC strains used, i.e. their ATCC designations, should be stated in the Materials and Methods section.
3. Lines 92-96: These two sentences should be written in the past tense.
4. Whilst the overall potency of MRS-2541 is high, there are some exceptions, e.g. VSE and *S. agalactiae*, that warrant further discussion. For example, do the authors hypothesize that the differences are target sequence-related? What are the implications of these observations in the context of the development of this potential antibacterial agent?
5. There are examples throughout the manuscript where genus and species names are incorrectly written. For example, on line 124 the word 'Staphylococci' does not need to be italicized and should be written with a lower case letter 's'. Likewise, on line 219 'Streptococci spp' should be written as 'streptococci'.
6. How many repeats of the MIC determinations were performed? Are the data presented in the tables single-point MIC values, or averages of, e.g., triplicate testing? Please provide this information in the 'Materials and Methods' section.
7. Line 272: 'is' should be 'are'.
8. Tables: Please clarify the difference between '-', 'NA' and '0.0' values in the 'S/I/R (%)' columns.
9. Table 2: The asterisk for the *E. faecium* / Daptomycin line does not appear to be defined.
10. Table 3: Is the lower MIC range measured for Viridans Group streptococci / penicillin combination written correctly? It states '{less than or equal to}0.0004', which is an order-of-magnitude less than the lower range for this agent against the other streptococci. Please check and confirm.

Reviewer #3 (Comments for the Author):

I have read, in furtherance of peer-review, the manuscript by Domagala et al entitled, *In vitro* activity of MRS-2541, a novel MetRS inhibitor, against a selection of 3 resistant Gram-positive organisms associated with serious hospital infections

This article appears to seek to expand upon a previously published study introducing a novel antibiotic, MRS-2541*, consequently, by some of the same authors. It is a synthesized inhibitor of methionyl t-RNA synthetase and therefore is an inhibitor of protein synthesis. Previous work indicated that it possessed specific activity against Gram-positive bacteria, and a high ratio of toxicity against mammalian cells.

The expansion that is sought by this manuscript is somewhat marginal. It looks to provide data on the performance of MRS-2541 on an array of drug-resistant, Gram-positive organisms that are commonly associated with healthcare-associated infections. None the less, the manuscript is largely well-constructed, written and contains a recognizably acceptable amount of data to support its conclusions. Additionally, data like that presented in this manuscript, which seek to expand the antibiotic armamentarium, is highly desirable to the biomedical realm. We are running out of antibiotics and we have precious few novel mechanisms at hand to develop new ones.

Below are individualized comments that warrant modifications of the manuscript. Should they be met forcefully and convincingly, the manuscript would be deemed acceptable to my experience and assessment.

1. The test panels that were used were of decent size (e.g. N=45 for *S. aureus*, 18; MSSA=18, MRSA=27, VISA=12....). What is not clear from the Materials and Methods or the Results/Discussion, is, what were the genetics of these tested isolates? Were they the same ones tested over and over again, or were they different strains? Different MLST? It would have made sense to test a diverse array of genomic subtypes of each category / species of organisms, but I can't seem to figure out what made up the multiples here.
2. The manuscript cuts very close to the previous manuscript, by many of the same authors. I think the intro should emphasize and better articulate the nature of the expansion of study that this work provides on top of the previous study. It feels like extra data was around and just wasn't added to the previous work. I think real expansion of knowledge occurred here, just make it clearer what that was.
3. It's probably beyond the scope of the manuscript to "wet lab" this suggestion but perhaps give merit to this in the Discussion? : What is expected to be the efficacy of killing against natural skin flora? Natural flora alterations lead to other health problems.
4. There appears to be an asterisk in Table 2, next to "6.7" in the S/I/R column. I don't see to what it refers.

Thank you.

*Molasky NMR, Zhang Z, Gillespie JR, Domagala J, Reyna D, Lipka E, Fan E, Buckner FS.2024.A novel methionyl-tRNA synthetase inhibitor targeting gram-positive bacterial pathogens. *Antimicrob Agents Chemother*68:e00745-24.<https://doi.org/10.1128/aac.00745-24>

I have read, in furtherance of peer-review, the manuscript by Domagala et al entitled,

In vitro activity of MRS-2541, a novel MetRS inhibitor, against a selection of 3 resistant Gram-positive organisms associated with serious hospital infections

This article appears to seek to expand upon a previously published study introducing a novel antibiotic, MRS-2541*, consequently, by some of the same authors. It is a synthesized inhibitor of methionyl t-RNA synthetase and therefore is an inhibitor of protein synthesis. Previous work indicated that it possessed specific activity against Gram-positive bacteria, and a high ratio of toxicity against mammalian cells.

The expansion that is sought by this manuscript is somewhat marginal. It looks to provide data on the performance of MRS-2541 on an array of drug-resistant, Gram-positive organisms that are commonly associated with healthcare-associated infections. None the less, the manuscript is largely well-constructed, written and contains a recognizably acceptable amount of data to support its conclusions. Additionally, data like that presented in this manuscript, which seek to expand the antibiotic armamentarium, is highly desirable to the biomedical realm. We are running out of antibiotics and we have precious few novel mechanisms at hand to develop new ones.

Below are individualized comments that warrant modifications of the manuscript. Should they be met forcefully and convincingly, the manuscript would be deemed acceptable to my experience and assessment.

1. The test panels that were used were of decent size (e.g. N=45 for *S. aureus*, 18; MSSA=18, MRSA=27, VISA=12....). What is not clear from the Materials and Methods or the Results/Discussion, is, what were the genetics of these tested isolates? Were they the same ones tested over and over again, or were they different strains? Different MLST? It would have made sense to test a diverse array of genomic subtypes of each category / species of organisms, but I can't seem to figure out what made up the multiples here.
2. The manuscript cuts very close to the previous manuscript, by many of the same authors. I think the intro should emphasize and better articulate the nature of the expansion of study that this work provides on top of the previous study. It feels like extra data was around and just wasn't added to the previous work. I think real expansion of knowledge occurred here, just make it clearer what that was.
3. It's probably beyond the scope of the manuscript to "wet lab" this suggestion but perhaps give merit to this in the Discussion? : What is expected to be the efficacy of killing against natural skin flora? Natural flora alterations lead to other health problems.
4. There appears to be an asterisk in Table 2, next to "6.7" in the S/I/R column. I don't see to what it refers.

Thank you.

*Molasky NMR, Zhang Z, Gillespie JR, Domagala J, Reyna D, Lipka E, Fan E, Buckner FS.2024.A novel methionyl-tRNA synthetase inhibitor targeting gram-positive bacterial pathogens. *Antimicrob Agents Chemother*68:e00745-24.<https://doi.org/10.1128/aac.00745-24>

February 4, 2026

Response to Reviewers for: Spectrum03615-25 (Domagala et al.)

1. Figure 1 is unnecessary because the chemical structure of MRS-2541 has previously been published.

Figure 1 has been eliminated

2. MIC data for the QC strains should be included as a separate line listing in the tables so that readers are assured the antibacterial susceptibility assay is performing within the expected (CLSI) guidelines. Also, the specific identities of the QC strains used, i.e. their ATCC designations, should be stated in the Materials and Methods section.

We have included the lines listings in Supplemental Tables 1, 2, and 3. The isolates were either independent non-consecutive clinical isolates from Microbiologics (MMX), MMX clinical isolates with specific susceptibility phenotypes, CDC or NARSA isolates with specific susceptibility phenotypes, or ATCC type or QC strains. The text has been updated with the following:

“Test organisms consisted of reference strains from the American Type Culture Collection (ATCC, Manassas, VA), the Centers for Disease Control and Prevention (CDC; Atlanta, CG), the Network on Antimicrobial Resistance in *Staphylococcus aureus* (NARSA; (BEI Resources, managed by ATCC), and independent non-consecutive clinical isolates from the Microbiologics repository (MMX, Kalamazoo, MI). The CDC, NARSA, and a subset of the MMX isolates were included to select for particular underrepresented susceptibility phenotypes. Quality control (QC) isolates including *S. aureus* ATCC 29213, *E. faecalis* ATCC 29212, and *S. pneumoniae* ATCC 49619 were included as necessary during testing (19, 20).”

3. Lines 92-96: These two sentences should be written in the past tense.

The tense has been corrected

4. Whilst the overall potency of MRS-2541 is high, there are some exceptions, e.g. VSE and *S. agalactiae*, that warrant further discussion. For example, do the authors hypothesize that the differences are target sequence-related? What are the implications of these observations in the context of the development of this potential antibacterial agent?

We have added to the discussion section a rationale for the weaker activity of *S. agalactiae*, *S. mitis*, and *Enterococcus faecium*. We have also added references to support our interpretations.

5. There are examples throughout the manuscript where genus and species names are incorrectly written. For example, on line 124 the word 'Staphylococci' does not need to be italicized and should be written with a lower case letter 's'. Likewise, on line 219 'Streptococci spp' should be written as 'streptococci'.

We have corrected the writing of genus and species names and thank the reviewers for pointing these out.

6. How many repeats of the MIC determinations were performed? Are the data presented in the tables single-point MIC values, or averages of, e.g., triplicate testing? Please provide this information in the 'Materials and Methods' section.

MIC values were determined for each isolate and drug combination in singleton. The methods have been updated.

7. Line 272: 'is' should be 'are'.

Correction made

8. Tables: Please clarify the difference between '-', 'NA' and '0.0' values in the 'S/I/R (%)' columns.

We thank the reviewer for pointing out this confusion. The “-“means that interpretive criteria do not exist. The one “NA” has been changed to “-“. 0.0 means no isolates have that interpretation. The footnote has been updated with this information.

9. Table 2: The asterisk for the *E. faecium* / Daptomycin line does not appear to be defined.

Thank you to the reviewer for catching this. There appeared to be an issue with that cell. The asterisk refers to this isolate having a susceptible-dose dependent breakpoint. The cell and footnote have been updated.

10. Table 3: Is the lower MIC range measured for Viridans Group streptococci / penicillin combination written correctly? It states '{less than or equal to}0.0004', which is an order-of-magnitude less than the lower range for this agent against the other streptococci. Please check and confirm.

Thank you for pointing out this error. It has been updated as “0.004”.

Reviewer #3 (Comments for the Author):

I have read, in furtherance of peer-review, the manuscript by Domagala et al entitled, In vitro activity of MRS-2541, a novel MetRS inhibitor, against a selection of 3 resistant Gram-positive organisms associated with serious hospital infections

This article appears to seek to expand upon a previously published study introducing a novel antibiotic, MRS-2541*, consequently, by some of the same authors. It is a synthesized inhibitor of methionyl t-RNA synthetase and therefore is an inhibitor of protein synthesis. Previous work indicated that it possessed specific activity against Gram-positive bacteria, and a high ratio of toxicity against mammalian cells.

The expansion that is sought by this manuscript is somewhat marginal. It looks to provide data on the performance of MRS-2541 on an array of drug-resistant, Gram-positive organisms that are commonly associated with healthcare-associated infections. None the less, the manuscript is largely well-constructed, written and contains a recognizably acceptable amount of data to support its conclusions. Additionally, data like that presented in this manuscript, which seek to expand the antibiotic armamentarium, is highly desirable to the biomedical realm. We are running out of antibiotics and we have precious few novel mechanisms at hand to develop new ones. Below are individualized comments that warrant modifications of the manuscript. Should they be met forcefully and convincingly, the manuscript would be deemed acceptable to my experience and assessment.

1. The test panels that were used were of decent size (e.g. N=45 for *S. aureus*, 18; MSSA=18, MRSA=27, VISA=12...). What is not clear from the Materials and Methods or the Results/Discussion, is, what were the genetics of these tested isolates? Were they the same ones tested over and over again, or were they different strains? Different MLST? It would have made sense to test a diverse array of genomic subtypes of each category / species of organisms, but I can't seem to figure out what made up the multiples here.

We have included the lines listings in Supplemental Tables 1, 2, and 3. The isolates were either independent non-consecutive clinical isolates from Microbiologics (MMX), MMX clinical isolates with specific susceptibility phenotypes, NARSA isolates with specific susceptibility phenotypes, or ATCC type or QC strains. The text has been updated with the following:

“Test organisms consisted of reference strains from the American Type Culture Collection (ATCC, Manassas, VA), the Centers for Disease Control and Prevention (CDC; Atlanta, CG), the Network on Antimicrobial Resistance in *Staphylococcus aureus* (NARSA; (BEI Resources, managed by ATCC), and independent non-consecutive clinical isolates from the Microbiologics repository (MMX, Kalamazoo, MI). The CDC, NARSA, and a subset of the MMX isolates were included to select for particular underrepresented susceptibility phenotypes. Quality control (QC) isolates including *S. aureus* ATCC 29213, *E. faecalis* ATCC 29212, and *S. pneumoniae* ATCC 49619 were included as necessary during testing (19, 20).”

2. The manuscript cuts very close to the previous manuscript, by many of the same authors. I think the intro should emphasize and better articulate the nature of the expansion of study that this work provides on top of the previous study. It feels like extra data was around and just wasn't added to the previous work. I think real expansion of knowledge occurred here, just make it clearer what that was.

We have revised the introductory text to reflect the relevance of this work relative to the current literature, in particular the literature on previous MetRS inhibitors. The text now reads:

“While the initial MICs vs laboratory strains indicated the potential for broad Gram-positive activity, we desired to examine a larger panel of clinical isolates biased with resistant phenotypes, which though rare in the general population provide valuable information on the full spectrum of MRS-2541. Most of the previous MetRS inhibitor literature (4A, 7A, 18) accentuated the excellent *S. aureus* and MRSA activity, but for the first time, we also want to report on a full panel of staphylococci species as well as MDR streptococci and enterococci given their growing role in Gram-positive hospital infections (9,10).”

3. It's probably beyond the scope of the manuscript to "wet lab" this suggestion but perhaps give merit to this in the Discussion? : What is expected to be the efficacy of killing against natural skin flora? Natural flora alterations lead to other health problems.

We feel that it is too speculative to discuss potential health effects of MRS-2541 on skin flora, although it is evident the compound has activity on *Staphylococcus epidermidis* (Table). This concern will require additional study outside the scope of this paper. Fortunately, the compound is not active on Gram(-) bacteria, thus the effects on the gut microbiome should be less than quinolones, cephalosporins, etc.

4. There appears to be an asterisk in Table 2, next to "6.7" in the S/I/R column. I don't see to what it refers.

Thank you to the reviewer for catching this. There appeared to be an issue with that cell. The asterisk refers to this isolate having a susceptible-dose dependent breakpoint. The cell and footnote have been updated.

Re: Spectrum03615-25R1 (In vitro activity of MRS-2541, a novel MetRS inhibitor, against a selection of resistant Gram-positive organisms associated with serious hospital infections)

Dear Prof. Frederick Buckner:

Your manuscript has been accepted, and I am forwarding it to the ASM production staff for publication. Your paper will first be checked to make sure all elements meet the technical requirements. ASM staff will contact you if anything needs to be revised before copyediting and production can begin. Otherwise, you will be notified when your proofs are ready to be viewed.

Sincerely,
Mark Pandori
Editor
Microbiology Spectrum